# BERT-AL: BERT FOR ARBITRARILY LONG DOCUMENT UNDERSTANDING

## ABSTRACT

Pretrained language models attract lots of attentions, and they take advantage of the two-stages training process: pretraining on huge corpus and finetuning on specific tasks. Thereinto, BERT (Devlin et al., 2019) is a Transformer (Vaswani et al., 2017) based model and has been the state-of-the-art for many kinds of Nature Language Processing (NLP) tasks. However, BERT cannot take text longer than the maximum length as input since the maximum length is predefined during pretraining. When we apply BERT to long text tasks, e.g., document-level text summarization: 1) Truncating inputs by the maximum sequence length will decrease performance, since the model cannot capture long dependency and global information ranging the whole document. 2) Extending the maximum length requires re-pretraining which will cost a mass of time and computing resources. Whats even worse is that the computational complexity will increase quadratically with the length, which will result in an unacceptable training time. To resolve these problems, we propose to apply Transformer to only model local dependency and recurrently capture long dependency by inserting multi-channel LSTM into each layer of BERT. The proposed model is named as BERT-AL (BERT for Arbitrarily Long Document Understanding) and it can accept arbitrarily long input without re-pretraining from scratch. We demonstrate BERT-ALs effectiveness on text summarization by conducting experiments on the CNN/Daily Mail dataset. Furthermore, our method can be adapted to other Transformer based models, e.g., XLNet (Yang et al., 2019) and RoBERTa (Liu et al., 2019), for various NLP tasks with long text.

## 1 INTRODUCTION

In recent years, neural networks are proposed to solve various NLP tasks. Especially, pretrained language models (Peters et al., 2018; Radford et al., 2018; Devlin et al., 2019; Yang et al., 2019) attract lots of attentions, which take advantage of the two-stages training process: pretraining on unlabeled corpus and then finetuning on specific tasks. The most famous model is BERT. BERT and its varieties are the state-of-the-art for many kinds of NLP tasks. The power of BERT does not only come from its architecture of networks, but also because it can be pretrained on a mass of text as a masked language model.

BERT can be used to solve almost all NLP tasks, and especially it can perform best on datasets with short text, e.g., GLUE (Wang et al., 2018) and Squad (Rajpurkar et al., 2016). However, there are still many document-level tasks, e.g., document-level text summarization (Hermann et al., 2015), long-document machine reading comprehension (Hewlett et al., 2016) and long text classification (Zhang et al., 2015). BERT cannot be finetuned for such tasks with long text directly or perform good on these tasks, since it is limited by the fixed-length position embedding which was determined during pretraining. We employ document-level text summarization as an example, which usually has longer text than the maximum sequence length of BERT.

Intuitively, there are two alternative solutions: 1) Truncating inputs by the maximum sequence length to fit the BERTs constraint. 2) Increasing the length of position embedding and re-pretraining the BERT from scratch. The first method will decrease performance, obviously, since some useful information placing behind the maximum sequence length is discarded by truncating. E.g., for text summarization, if the key point sentence locates at the end of the document, it never be recalled

even though the model is powerful. For the second method, re-pretraining the BERT from scratch will cost a mass of computing time and resources. Whats even worse is that the computational complexity will increase quadratically with the length, which will result in an unacceptable training time. For example, the XLNet-Large (Yang et al., 2019) costs 2.5 days on 512 TPU v3 chips and the RoBERTa (Liu et al., 2019) costs 1 day on 1024 V100 GPUs.

To resolve these problems, we propose BERT-AL (BERT for Arbitrarily Long Document Understanding) model that extracts local features by applying parallel multi-layer Transformers into chunked input and employs multi-channel LSTMs to capture global information crossing Transformers. This fusion breaks the limitation of BERT by the ability of capture unlimited sequential information from LSTM, and makes it be able to process arbitrarily long text. On the other hand, it also skillfully avoids gradient vanishing and exploding problem (Li et al., 2018) of LSTM since only few steps are required by multi-channel LSTM. Therefore, BERT-AL can solve the problems of original BERT: 1) For document-level tasks, BERT-AL can directly take all text as the input without truncating. 2) When the input length is longer then maximum sequence length of BERT, BERT-AL still can load the pretrained BERT model, which avoids pretraining much longer BERT model from scratch.

We demonstrate BERT-ALs effectiveness on text summarization by conducting experiments on the CNN/Daily Mail dataset (Hermann et al., 2015) with various maximum sequence lengths of pretrained BERT. The results prove that BERT-AL can consistently outperform BERTSUM (Liu, 2019) which is the BERT-based state-of-the-art, when finetuning from the pretrained BERT model with the same maximum sequence length. Additionally, BERT-AL is a general NLP model which has no specific setting for text summarization, so it can be easily adapted to other tasks with long text, e.g., document-level machine reading comprehension and long text classification. Furthermore, the method, applying multi-channel LSTM on hidden states from transformers, also can be used in other Transformer based pretrained models, e.g., XLNet and RoBERTa.

In summary, contributions of this paper are shown as follow.

1) We propose a new architecture that combines Transformer and LSTM, which resolve the problem Transformer cannot be used in very long text, and skillfully avoid LSTMs drop backs.

2) We propose multi-channel LSTM only applied on the corresponding position across different Transformers, which can take fully advantage of LSTM without hurting Transformers representation too much.

3) We conduct experiments to prove BERT-AL can outperform other models with the BERT pretrained under the same maximum sequence length, and can perform very close to BERTSUM with at least twice the maximum length than ours.

## 2 BACKGROUND

### 2.1 BERT

According to the original implementation described in Vaswani et al. (2017), BERT is a multi-layer bidirectional Transformer encoder, and Multi-Head Self-Attention is the key structure of Transformer encoder.

For the input $H \in \mathbb{R}^{L \times D}$ of each Transformer layer, $H$ will be mapped to three different spaces, named as $Q$, $K$ and $V$, respectively. Self-attention computes the dot products between $Q$ and $K$ to roughly get the weight matrix, and then multiply $V$ to get the hidden representation. Multi-head mechanism promotes the power of Transformer because it allows the model to jointly attend to information from different representation subspaces at different positions.

BERT employs two tasks for pretraining: Masked Language Model and Next Sentence Prediction.

**Masked Language Model**: BERT proposes a bidirected language model, which replaces 15% words in text by $[MASK]$ label and then predicts which words they are. To mitigate the mismatch between pretraining and finetuning, i.e., the $[MASK]$ token does not appear during finetuning, they do not always replace masked words with the actual $[MASK]$ token. Then, BERT will predict the original token with cross entropy loss.

**Next Sentence Prediction**: BERT is also pretrained by the next sentence prediction task. Specifically, when choosing the sentences A and B for each pretraining example, 50% of the time B is the actual next sentence that follows A (labeled as $IsNext$), and 50% of the time it is a random sentence from the corpus (labeled as $NotNext$). Then, BERT will predict whether the sentence B is the actual next sentence that follows A or not with cross entropy loss.

For most of all NLP tasks, BERT concatenates different parts of input into a sequence beginning with $[CLS]$ token and inserts $[SEP]$ token between two different parts. Before going through Transformer layers, BERT merges three different embeddings into one, i.e., word embedding, position embedding and segment embedding. Thereinto, position embedding is the information about the relative or absolute position of the tokens in the sequence. Since the model contains no recurrence and no convolution, position embedding is added to make use of the order of the sequence. There are two implements of position embedding, i.e., learned position embedding and sine and cosine functions of different frequencies, but both of them cannot expand to longer without a performance regression (Wang et al., 2019a).

## 2.2 BERTSUM

BERTSUM is an extension of BERT on extractive text summarization task and it truncates only the first 512 tokens as input. To select sentences, BERTSUM adds $[CLS]$ to the head of each sentence and $[SEP]$ to the tail of each sentence indicating the end of that sentence. The following summarization layer will score each $[CLS]$ which presents the importance of that sentence. Finally, sentences with top 3 highest scores compose the summary. There are three summarization layers proposed in Liu (2019):

**1) Simple Classifier:** only adds a linear layer on each $[CLS]$ and uses a sigmoid function to get the predicted score:

$$\hat{Y}_i = \sigma(W_o T_i + b_o) \tag{1}$$

where $T_i$ is the logit of $i$th sentence, $\sigma$ is the Sigmoid function.

**2) Inter-Sentence Transformer:** applies more Transformer layers into sentence-level representations as follows.

$$\begin{aligned}
\tilde{h}^l &= LN(h^{l-1} + MHAtt(h^{l-1})) \\
h^l &= LN(\tilde{h}^l + FFN(\tilde{h}^l)) \\
\hat{Y}_i &= \sigma(W_o h_i + b_o)
\end{aligned} \tag{2}$$

where $h^0 = PosEmb(T)$ and $T$ are the sentence vectors output by BERT, $PosEmb$ is the function of adding position embeddings to $T$; $LN$ is the layer normalization operation; $FFN$ is a feedforward network; $MHAtt$ is the multi-head attention operation; the superscript $l$ indicates the depth of the stacked layer. The final output layer is still a sigmoid classifier and $T_i$ is the logit of $i$th sentence.

**3) Recurrent Neural Network:** applies an LSTM layer over the BERT outputs to learn summarization-specific features. At time step $i$, the input to the LSTM layer is the BERT output $T_i$, and the output is calculated as:

$$\begin{aligned}
\begin{pmatrix} F_i \\ I_i \\ O_i \\ G_i \end{pmatrix} &= LN_h(W_h h_{i-1} + LN_x(W_x T_i)) \\
C_i &= \sigma(F_i \odot C_{i-1}) + \sigma(I_i) \odot tanh(G_{i-1}) \\
h_i &= \sigma(O_t) \odot tanh(LN_c(C_t)) \\
\hat{Y}_i &= \sigma(W_o h_i + b_o)
\end{aligned} \tag{3}$$

where $F_i, I_i, O_i$ are forget gates, input gates, output gates; $G_i$ is the hidden vector and $C_i$ is the memory vector; $h_i$ is the output vector; $LN_h, LN_x, LN_c$ are there difference layer normalization operations; The final output layer is also a sigmoid classifier and $T_i$ is the logit of $i$th sentence.

However, we argue that the way of truncation will result in the loss of information in the latter part of the document. If the key sentences locate at the end of the document, they will never be recalled

even though the model is powerful. In contrast, if model can take all the tokens of the document as input, it will get these key sentences and produce better result. Therefore, we propose BERT-AL to solve this problem. For simplicity, we only employ the Simple Classifier as the summarization layer in the following sections.

## 3 BERT-AL

In this section, we will introduce the detail about BERT-AL, and as illustrated in Figure 1, BERT-AL has mainly two key components different with BERTSUM: one is the multi-channel LSTM and another is the positional encoding.

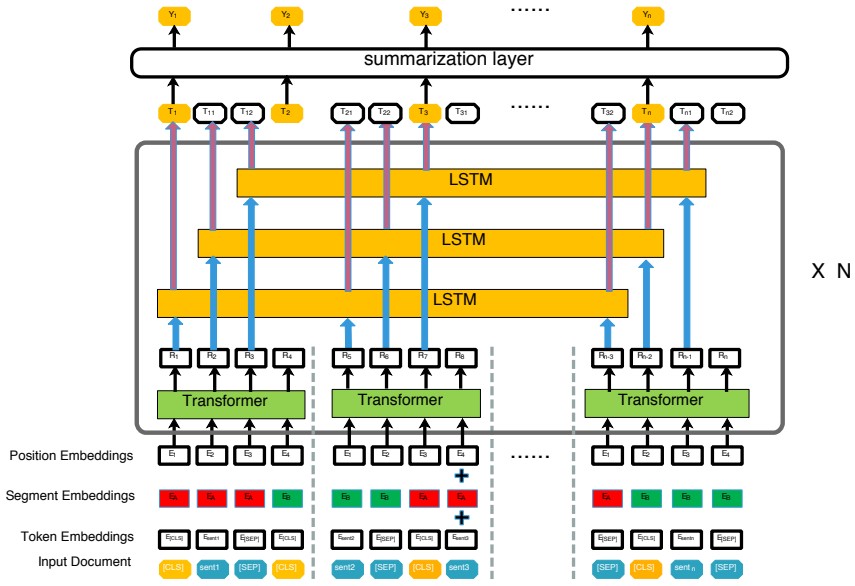

Figure 1: The architecture of BERT-AL

### 3.1 MULTI-CHANNEL LSTM

We first give the definition of our scenario: assume that the length of the document is $l_{doc}$, and we have only a pretrained BERT model with the maximum sequence length $l_{BERT}$. We first split the document into $n_{segment}$ segments and each segment has the length $l_{BERT}$ (the length of $n_{segment}$th segment is shorter than $l_{BERT}$). We set $l_{BERT-AL} = n_{segment} * l_{BERT}$ as the maximum sequence length of BERT-AL, and then we have $l_{BERT-AL} \geq l_{doc}$, which means BERT-AL can take arbitrarily long document as input.

As we know, LSTM has the capability of capturing information across arbitrarily long steps but is weak in capturing long-term dependencies. Therefore, within a segment, we use Transformer to capture long-term dependencies and extract the feature via self-attention among local positions, and then fully represent the segment at each layer. To this end, we propose a multi-channel LSTM to chain the representation of different segments. For the above definition, $l_{doc}$ can be arbitrarily long and so is $n_{segment}$. Thus, we apply LSTM on $n_{segment}$ dimension, and the steps of LSTM should also be $n_{segment}$.

As Figure 1 shows, LSTM is following the segment-wise Transformers at each layer and takes the hidden states from Transformers as its input. To fit the next layer, we set LSTM's hidden size also as $d_{model}$, i.e., 768 for BERT-Base and 1024 for BERT-Large. Thus, the computing in each layer is shown as follows.

$$H_{i-1} = Concat(H_{i-1,1}, H_{i-1,2}, ..., H_{i-1,n_{segment}})$$
$$H_{i,j}^{Trans} = Transformer(H_{i-1,j} \in R^{[l_{BERT}, d_{model}]}) \qquad (4)$$
$$H_i = LSTM(H_i^{Trans} \in R^{[l_{BERT}, n_{segment}, d_{model}]})$$

where $H_i$ is the hidden states from $i$th layer, $H_{i,j}$ is $j$th segment in $H_i$, and $H_{i,j}^{Trans}$ is the hidden states from $j$th Transformer. All segment-wise Transformers at the same layer share parameters.

BERT-AL employs a single layer of unidirectional LSTM as the implement of multi-channel LSTM, and a channel is corresponding to a relative position within a segment. Per-gate layer normalization is applied in each LSTM cell (Ba et al., 2016). We do not split the segments according to natural sentences, so $[CLS]$ may be at the beginning, in the middle or at the end of a block. Similarly, the output at $[CLS]$ positions also do not come from the same LSTM channel. The reason we apply multi-channel LSTM on $n_{segment}$ rather than a 2D LSTM on the whole sequence is taking advantage of LSTM's expandability on variable-length segments, but meanwhile, reducing the interference to self-attention among positions within a segment.

## 3.2 EMBEDDING

BERT-AL take the same input format as BERTSUM does, i.e., adding $[CLS]$ to the head of each sentence and $[SEP]$ to the tail of each sentence. Since BERT-AL has multiple segment-wise Transformers at the first layer, its input embedding is also changed. More details are shown as follows.

Before feeding embedding to Transformers, the sequence is divided into segments whose lengths are the same with the pretrained BERT model's maximum sequence length. Token embedding and segment embedding are also same as the original BERT. For position embedding, we also reserve the learned position embedding used in the original BERT. However, if we do not re-pretrain the BERT, the dimension of position embedding matrix in the pretrained BERT model is $[l_{BERT}, d_{model}]$, while the length of task's input is $l_{BERT-AL}$. To resolve this problem, we copy the original position embedding matrix $n_{segment}$ times and concatenate them. Therefore, we can get a position embedding matrix with dimension $[l_{BERT-AL}, d_{model}]$. Through this way, each position in the input will get a positional encoding. Therefore, the model can work correctly. In the Transformer layer, each segment does not interact with each other. Therefore, although the position embeddings are same in different segments, it does not affect the model to extract feature in its own segment.

## 3.3 OTHER DETAILS

It is obvious that each sentence's length is different, but segments have the same length. Therefore, the start of each segment may not be $[CLS]$ and the tail may not be $[SEP]$, either. In Figure 1, the $T_i$ is the representation of $i$ sentence through $N$ Transformers and LSTMs. After summarization layer, $Y_i$ represents the score of that sentence, and then we choose sentences with top 3 highest scores compose the summary. Finally, Table 1 shows difference of BERT-AL and BERTSUM.

Table 1: The difference of BERT-AL and BERT-SUM

|  | BERTSUM | BERT-AL |
|---|---|---|
| Architeture | BERT + summarization layer | segment-wise BERT + multi-channel LSTM + summarization layer |
| Position embedding | same as BERTs position embedding | copy and concatenate BERTs position embedding |
| Max input length | BERTs input length | arbitrarily long |

## 4 EXPERIMENT

In this section, we demonstrate BERT-ALs effectiveness on text summarization by conducting experiments on the CNN/Daily Mail dataset. We compare BERTSUM with our models on various different settings, since BERTSUM is the state-of-the-art on CNN/Daily Mail dataset.

### 4.1 CNN/DAILY MAIL DATASET

The CNN/Daily Mail dataset contains news articles and associated highlights, i.e., a few bullet points giving a brief overview of the article. We used the standard splits as training, validation and testing sets as BERTSUM did. The statistics of datasets is shown in Tabel 2.

Table 2: Statistics of CNN/Daily Mail dataset

|          | CNN   | Daily Mail | Total  |
|----------|-------|------------|--------|
| Train    | 90266 | 196961     | 287227 |
| Validate | 1220  | 12148      | 13368  |
| Test     | 1093  | 10397      | 11490  |

We also perform the same preprocessing for data as BERTSUM did, including keeping entities, splitting sentences by CoreNLP and following methods in See et al. (2017).

We employ ROUGE-1, 2, L (Lin, 2004) to evaluate the performance for different methods and focus on the $F_1$ score, as follows.

$$R_{ROUGE-N} = \frac{\sum_{S \in \{ReferenceSummaries\}} \sum_{gram_n \in S} Count_{match}(gram_n)}{\sum_{S \in \{ReferenceSummaries\}} \sum_{gram_n \in S} Count(gram_n)}$$

$$P_{ROUGE-N} = \frac{\sum_{S \in \{ReferenceSummaries\}} \sum_{gram_n \in S} Count_{match}(gram_n)}{\sum_{S \in \{CandidateSummaries\}} \sum_{gram_n \in S} Count(gram_n)} \tag{5}$$

$$F_{ROUGE-N} = \frac{(1+\beta^2)R_{ROUGE-N}P_{ROUGE-N}}{R_{ROUGE-N} + \beta^2 P_{ROUGE-N}}$$

where $n$ stands for the length of the n-gram, $gram_n$, and $Count_{match}(gram_n)$ is the maximum number of n-grams co-occurring in a candidate summary and a set of reference summaries.

$$R_{lcs} = \frac{LCS(X,Y)}{m}$$

$$P_{lcs} = \frac{LCS(X,Y)}{n} \tag{6}$$

$$F_{lcs} = \frac{(1+\beta^2)R_{lcs}P_{lcs}}{R_{lcs} + \beta^2 P_{lcs}}$$

where $X$ is a reference summary sentence and $Y$ is a candidate summary sentence, $m$ is the length of $X$ and $n$ is the length of $Y$, $LCS(X,Y)$ is the length of a longest common subsequence of $X$ and $Y$.

To adapt dataset to suit extractive summarization task, we also use a greedy algorithm which is same as BERTSUM to generate an oracle summary for each document. The algorithm selects a set of sentences as the oracle set, which can maximize the ROUGE scores.

## 4.2 EXPERIMENT SETTING

Table 3: Setting of experiment groups

|         | $l_{BERT}$ | $l_{BERT-AL}$ | $n_{segment}$ |
|---------|------------|---------------|---------------|
| Group-1 | 8          | 512           | 64            |
| Group-2 | 16         | 512           | 32            |
| Group-3 | 128        | 512           | 4             |
| Group-4 | 256        | 1024          | 4             |

Assume that we have only a pretrained BERT model and the length of its position embedding is $l_{BERT}$. Our task is to produce summaries on documents with $l_{doc}$ tokens. The BERT-AL can take $l_{BERT-AL}$ length tokens as input ($l_{BERT-AL} = n_{segment} * l_{BERT}$ and $l_{BERT-AL} >= l_{doc}$). We design four groups of experiments and set $l_{BERT} = 8, 16, 128, 256$ for them. More details about settings are shown in Table 3. Under each group setting, we compare three different methods, including Baseline-1, Baseline-2 and BERT-AL. More details about methods are shown as follows.

1) **Baseline-1** applies BERTSUM to the dataset with length = $l_{BERT}$, and then truncates text longer than $l_{BERT}$.

2) **Baseline-2** concatenates the first $l_{BERT}$ position embedding of the original BERT with a randomly initialized embedding with the length $(n_{segment} - 1) * l_{BERT}$, and then employs the concatenated embedding as BERTSUM's position embedding. Finally, it applies BERTSUM to long documents with length $= l_{BERT-AL}$.

3) **BERT-AL** copies the position embedding matrix $n_{segment}$ times and concatenates them along the dimension of length, and then employs the embedding as the position embedding of segment-wise Transformers at corresponding layers. Finally, it applies BERT-AL to long documents with length $= l_{BERT-AL}$

The Baseline-1 aims to compare BERTSUM and BERT-AL while the Baseline-2 aims to reveal the effectiveness of multi-channel LSTM. Furthermore, Baseline-2 can be used to indicate whether the promotion of BERT-AL is due to the longer input or the multi-channel LSTM.

For fairly comparison, we set all hyperparameters equal to BERTSUM reported in Liu (2019). Specifically, we set the number of layers as 12, the hidden size as 768, the number of self-attention heads as 12 and the feed-forward size as 3072. We use Adam with $\beta_1 = 0.9$, $\beta_2 = 0.999$ and $\epsilon = 10^{-9}$. We also use a linear learning rate decreasing scheduler with warming-up on first 10,000 steps. All models are trained for 50,000 steps with gradient accumulation per two steps. We also select the top-3 checkpoints based on their evaluation losses on the validations set and report the averaged results on the test set.

## 4.3 RESULTS AND ANALYSIS

The results are showed in Table 4, including BERTSUM[1]. We can obtain the following observations:

Table 4: Experiment results

|  | Model | ROUGE-1 | ROUGE-2 | ROUGE-L |
|---|---|---|---|---|
|  | BERTSUM | 42.94 | 20.14 | 39.38 |
| Group-1 | Baseline-1 | 6.47 | 1.24 | 5.68 |
|  | Baseline-2 | 40.38 | 17.86 | 36.77 |
|  | **BERT-AL** | **41.26** | **18.63** | **37.69** |
| Group-2 | Baseline-1 | 11.81 | 3.69 | 10.47 |
|  | Baseline-2 | 40.44 | 17.93 | 36.83 |
|  | **BERT-AL** | **41.73** | **19.00** | **38.18** |
| Group-3 | Baseline-1 | 41.19 | 18.57 | 37.50 |
|  | Baseline-2 | 41.44 | 18.77 | 37.83 |
|  | **BERT-AL** | **42.14** | **19.38** | **38.60** |
| Group-4 | Baseline-1 | 42.30 | 19.58 | 38.73 |
|  | Baseline-2 | 42.27 | 19.57 | 38.72 |
|  | **BERT-AL** | **42.61** | **19.79** | **39.07** |

1) For all of the four groups, BERT-AL outperforms baselines, consistently. It proves that BERT-AL is effective on long document summarization task, which comes from merging Transformer's local feature extraction ability and LSTM's global time capturing ability.

2) Comparing Baseline-1 across four groups, performance increases with $l_{BERT}$ is longer. It implies longer input contains more useful information and truncating input leads to a performance drop.

3) Comparing Baseline-1 with Baseline-2, Baseline-2 performs better than Baseline-1 under nearly all settings. It implies randomly initialized position embedding also can help capture longer information even only with finetuning. However, this improvement will decrease with the maximum sequence length increases, and Baseline-2 has a worse perform than Baseline-1 when $l_{BERT} = 256$.

4) Comparing BERT-AL and Baseline-1 in Group-4, BERT-AL can further promote the ROUGE score and break up the bottleneck of Baseline-2 when the $l_{BERT}$ is large. It implies repeated position embedding with multi-channel LSTM is more effective than random initialized one.

---

[1]The result comes from our reproduction

5) Comparing Baseline-2 with BERT-AL in all four groups, the promotion of ROUGE score is more significant when the number of *segments* (i.e., LSTM's time-steps) is larger. It implies that the LSTM's capability of timing capture is not be fully utilized when the LSTM has less time-steps.

## 4.4 DISCUSSION

BERT-AL is designed as the model with a shorter pretrained BERT, and it still can achieve a comparable performance to BERTSUM. In Group 4 experiments, we can find BERT-AL (256)s performance has been very closed to BERTSUM (512). Simultaneously, BERT-AL has a much faster training and inference speed, since Transformers runtime is proportional to $l^2$ and multi-channel LSTM is high parallelly applied on $n_{segment}$ steps. Therefore, BERT-AL can be a good alternative under the following situations: 1) For a NLP task, the input text is too long to feed into a BERT model due to GPU memory or other limitations. 2) The time of pretraining a longer model from scratch is unacceptable under a limited computing resource. 3) There is restraint for inference speed, and then we also can split the text into small segments and feed them into BERT-AL.

## 5 RELATED WORK

There are several works related to modeling recurrence for self-attention network (SAN) in Transformer. Dehghani et al. (2018) recurrently refines the representations of each layer to improve SAN encoder. Shen et al. (2018) introduces a directional self-attention network (DiSAN), which only allows each token to attend previous (or following) tokens. Both Hao et al. (2019) and Chen et al. (2018) propose to combine SAN encoder with an additional RNN encoder. The former enhances the Transformer with recurrence information, while the latter augments RNN-based models with SAN encoder. Wang et al. (2019b) adds a local RNN layer in front of SAN in each Transformer layer, which aims to capture both local structures and global long-term dependencies in sequences. Wang et al. (2019a) adds LSTM after all the Transformer by the guide of Coordinate Architecture Search. However, these above works all target to solve the limitation of positional encoding, while our model aims to apply pretrained BERT to longer text and need not to pretrain from scratch, which reduces the cost of time and computing resources. Transformer-XL (Dai et al., 2019) contains segment-level recurrence with state reuse and relative positional encoding for language model beyond a fixed-length context, which is similar with our model and also can processes longer text. However, Transformer-XL aims at language model and need to pretrain the model from scratch.

For processing long input, Bengio et al. (2013) proposes "conditional computation" which is to only compute a subset of a networks units for a given input by gating different parts of the network. Ling & Rush (2017) proposes a coarse-to-fine attention model that uses hard attention to find the text chunks of importance and then only attend to words in that chunk. Liu et al. (2018) first coarsely selects a subset of the input, then trains an model while conditioning on this subset. Cohan et al. (2018) proposes a hierarchical encoder which represents each section using word-level RNN and then represents all sections using section-level RNN. These works is all similar with our method, which is splitting the long input into several subsets to process them respectively and then merging them. However, they are all need re-pretrain the model from scratch. Furthermore, we use the most powerful model–BERT.

## 6 CONCLUSION

BERT has been the state-of-the-art for all kinds of NLP tasks. However, BERT cannot be applied to long text tasks, e.g., document-level text summarization, because it cannot take text longer than the maximum length as input. However, the maximum length is predefined during pretraining, and expanding maximum length need re-pretrain which will cost lots of computing resource. We propose a novel model named BERT-AL which combines the advantages of Transformer and LSTM. The BERT-AL can take arbitrarily long text as its input and need not re-pretrain from scratch. We demonstrate BERT-ALs effectiveness on the text summarization task by conducting experiments on the CNN/Daily Mail dataset, and the experimental results prove that BERT-AL is effective on NLP tasks with very long text as input. Furthermore, our model can be easily adapted to various tasks and pretrained models.

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
