# OpenReview forum: "BERT-AL: BERT for Arbitrarily Long Document Understanding"
_ICLR.cc/2020/Conference — Reject_

### Official Review · AnonReviewer2 · 2019-10-19
**Official Blind Review #2**

**Rating:** 3

**Review:**

This paper proposed another variant of BERT, called BERT-AL, which can deal with arbitrarily long inputs. The authors constructed the proposed method by combining the segment-wise BERT with the multi-channel LSTM. The authors validated the proposed method on the text summarization task and achieved higher performance than existing works. Their proposed method can be combined with other transformer-based approaches, such as XLNet or RoBERTa.

The paper tackles a relevant topic that is interesting to the attendees of ICLR, which is, “How do we get the overall information of a document, which has a number of sentences?”.  However, the paper has several issues that I will try to cover in the following.

(1) (No sufficient results which support authors’ claim)
First: In the introduction, authors claim that the model for NLP tasks whose inputs are usually long texts should be able to take arbitrarily long inputs so that the model can extract important information from the entire long documents. However, there is no concrete experimental example, where the existing methods (standard BERT or BERTSUM) fail to extract information.
Second: In the discussion of the experiment section, the authors claim that the proposed method BERT-AL can be parallelized and runs faster than existing methods. However, there is no comparison with existing methods about the running time.

(2) (Clarity about the proposed method)
The presentation of the proposed method is slightly confusing. There is no concrete explanation about the learning procedure. Are any pre-training and fine-tuning procedures the same with those of the standard BERT?
The main part of the proposed method is to combine the segment-wise BERT with the multi-channel LSTM. How is the pre-training procedure conducted with parallelized computation? A detailed explanation should be shown.

(3) (Clarity about the experimental results)
In table 4, the experimental results are shown. There are results of the baseline 1 & 2 (existing methods), the proposed method, and the existing method (BERTSUM). There is no discussion about the comparison of the proposed method with BERTSUM. It seems that BERTSUM’s result is superior to the proposed method’s result. Please explain more details of those results.
The paper includes an explanation about the dataset and the task but does not include the experimental environment (used machines and the time to conduct the experiment). For the fair evaluation of results, we need the information on the overall experimental environment.

(4) (Presentation)
The presentation around mathematical formulas is a bit confusing. Please fix the following.
- Subscripts of the formula should be ‘\mathrm’
- ‘<=‘ should be ‘\le’
- The ’n’ of the right-hand side of equation (5) is ’N’?
- The multiplication ‘*’ should be ‘\times’

**Experience Assessment:**

I do not know much about this area.

**Review Assessment: Checking Correctness Of Derivations And Theory:**

N/A

**Review Assessment: Checking Correctness Of Experiments:**

I assessed the sensibility of the experiments.

**Review Assessment: Thoroughness In Paper Reading:**

I read the paper at least twice and used my best judgement in assessing the paper.

---

### Official Review · AnonReviewer4 · 2019-10-27
**Official Blind Review #4**

**Rating:** 3

**Review:**

This paper proposed a BERT based document summary model that has the capability of modeling arbitrarily long documents. Experiments on CNN/Daily dataset show the improvement of the proposed model compared with its baselines.
The advantage of this paper is that the time-consuming training process of BERT can be avoided.

I think this paper is not good enough to be accepted. The model does not have the ability of understanding “Arbitrarily” long documents, it just aggregates multiple (n_segment) BERT segments and extends the BERT capability from l_bert to n_segment*l_bert. It is not as flexible as the LSTM which can capture the real arbitrary long document. The model looks bloated and the performance is not persuasive. The authors reproduced the baseline BERTSUM but the reproduced performance is significantly lower than performance in the original BERTSUM paper. (without explanation) The proposed model can’t beat the performance in the original BERTSUM paper. Even to their reproduced BERTSUM results, the proposed model gives very closed performances.

Besides, this paper doesn’t compare their model with other SOTA models and all the experiments are conducted in one dataset. The result analysis is not enough. As the main contribution in this paper is to understand the longer documents, the performance on long documents should be evaluated separated.

**Experience Assessment:**

I have published in this field for several years.

**Review Assessment: Checking Correctness Of Derivations And Theory:**

I assessed the sensibility of the derivations and theory.

**Review Assessment: Checking Correctness Of Experiments:**

I carefully checked the experiments.

**Review Assessment: Thoroughness In Paper Reading:**

I read the paper at least twice and used my best judgement in assessing the paper.

---

### Official Review · AnonReviewer3 · 2019-10-30
**Official Blind Review #3**

**Rating:** 6

**Review:**

The author proposes an extended version of the BERT architecture, BERTAL for text summarization. BERTAL aims to overcome the limit of maximal allowable text length in BERT, and the experiments show some consistent performance enhancement to baseline architecture using BERT and approaching performance to state-of-the-art architecture using BERT (BERTSUM), where BERTSUM has the maximal length limit .
           The experimental procedures as well as the choice of architectural design are well explained and designed in a logical way. Substantial comparison experiments also pinpoint the performance.
            However, it doesn’t compare the result of BERTAL to other text summarizers with arbitrary-length text, which is not based on BERT.

**Experience Assessment:**

I do not know much about this area.

**Review Assessment: Checking Correctness Of Derivations And Theory:**

I assessed the sensibility of the derivations and theory.

**Review Assessment: Checking Correctness Of Experiments:**

I assessed the sensibility of the experiments.

**Review Assessment: Thoroughness In Paper Reading:**

I made a quick assessment of this paper.

---

### Official Review · AnonReviewer5 · 2019-11-04
**Official Blind Review #5**

**Rating:** 3

**Review:**

The paper proposes a methodology to overcome the problem of processing long sequences with a pre-trained Transformer model, which suffers from high computational costs due to the complexity being quadratic in the length of the sequence. The authors also point out that BERT needs to be retrained from scratch if sequences longer than the specified maximum length (512) are to be processed. Their method (BERT-AL) chunks the input text into segments of maximum length. Each segment is propagated through several layers of a pre-trained Transformer layer followed by a multi-channel LSTM. Herein, the positional embeddings in each segment are the same. The model is applied to extractive summarization, and directly compared to the BERTSUM model, which can only process documents up to length 512. The comparison is made for 4 application scenarios, each corresponding to an artificial maximum length after which the BERTSUM model truncates the input: after 8, 16, 128, and 256 tokens. The experimental results suggest that BERT-AL outperforms BERTSUM in all 4 scenarios: Substantially for max length 8 and 16, and marginally for 128 and 256. BERTSUM without any truncation still performs best, however.

I think the paper should be rejected for three main reasons: (1) The idea of using RNNs to overcome the long sequence problem in Transformers is already well studied. The specific proposed architecture may be new, but the design choices are not well justified. (2) The experimental evaluation is not convincing: The application scenarios are unrealistic, the dataset is not well-chosen, and the results are not impressive. (3) The structure and language of the paper needs to be polished.

Regarding (1): The problem of dealing with long sequences in Transformer models has been known for a long time, and there already exist several proposed solutions based on an RNN component, which are acknowledged in this paper. The authors propose another model for a similar purpose. While the model is interesting, the individual design decisions are not well-justified, e.g., why the LSTM is applied at each layer and why it needs to be a multi-channel LSTM. Neither a comparison with models from previous works nor an ablation study is provided, so that there is also no empirical justification for the model design.
The authors argue that the novelty in their model comes from its applicability to pre-trained models, e.g., BERT and XLNET. While the motivation for BERT is reasonable, it is questionable whether this will still be a relevant for future research, as XLNET already mitigates the problem through the reliance on Transformer-XL, which is conceptually very similar to BERT-AL. If future pretrained models account for long sequences already during pre-training, the motivation for this work is rather low.

Regarding (2): In order to show the superiority of BERT-AL over BERTSUM, it would've been natural to employ it to datasets whose data are beyond what standard BERT can handle, i.e., longer than 512 tokens. Instead, the authors chose to evaluate on 4 rather unrealistic application scenarios derived from the CNN/DailyMail dataset, which artificially limit the range of the pre-trained BERT model to much less than what it can actually handle. While I understand the idea behind this setup, the insight that BERT performs poorly if the text is truncated to 8 or 16 tokens is not helpful. Since the improvement of BERT-AL over the baselines becomes marginal for longer sequences, and the performance of BERTSUM, without truncating the text, is not reached, I am not convinced of the model's value. An evaluation on more suitable datasets could help here.

Regarding (3): The paper is difficult to read for two reasons. First, the paper is not structured in a way that facilitates the understanding of the proposed method. For example, it is not clear why the pre-training tasks of BERT (masked language modeling and next sentence prediction) are relevant enough to be so prominently described. If the method is applicable to many pre-trained Transformers (as was claimed), then there is no need to go into this level of detail. On the same note, there is no need to explain BERTSUM's Inter-Sentence Transformer and Recurrent Neural Network variants for half a page if they are dismissed in the next paragraph. Second, the paper contains a lot of typos and grammatical mistakes. This is not a deal-breaker by itself, but it makes it obvious that the paper needs substantial polishing before it should be published.

Questions and suggestions to the authors:

1) What is the advantage of choosing a multi-channel LSTM, where each segment is a channel? Wouldn't a single channel suffice that processes aggregated information from the segment (as the Transformer computes)? I think your paper would benefit from exploring different options like that experimentally to better justify your model decisions.

2) It is not clear to me how the information exchange between layers happens. If each layer consists of a single layer from a pre-trained Transformer followed by an LSTM, the input to the following layer of the pre-trained Transformer is the output of the LSTM, which is not part of the pre-trained model. That is, the following Transformer layer receives an input that is very different from what it has seen during pre-training, which we can not expect to work. If there is a misunderstanding on my side, could you please clarify that, or otherwise make it clearer in the paper?

3) I think a good baseline would be to do segment-wise encoding via the pre-trained Transformer, and then feed all sentence representations (i.e., the respective CLS tokens) from all segments into the Transformer-based summarization layer from BERTSUM. This would be a compromise between your model (doing segment-wise encoding but not using an LSTM) and BERTSUM, and thus could potentially highlight the importance of the LSTM.

4) I don't think showing the number of examples in your datasets as in Table 2 contributes much to the paper. Instead, I suggest to show statistics on the length of the documents / number of sentences etc., because these are directly relevant to your study.

**Experience Assessment:**

I have read many papers in this area.

**Review Assessment: Checking Correctness Of Derivations And Theory:**

N/A

**Review Assessment: Checking Correctness Of Experiments:**

I assessed the sensibility of the experiments.

**Review Assessment: Thoroughness In Paper Reading:**

I read the paper thoroughly.

---

### Decision · Program_Chairs · 2019-12-19

**Decision:**

Reject

**Comment:**

This paper proposes a hybrid LSTM-Transformer method to use pretrained Transformers like BERT that have a fixed maximum sequence lengths on texts longer than that limit.

The consensus of the reviewers is that the results aren't sufficient to justify the primary claims of the paper, and that—in addition—the missing details and ablations cast doubt on the reliability of those results. This is an interesting research direction, but substantial further experimental work would be needed to turn this into something that's ready for publication at a top venue.